

# A peptide encoded by the circular form of the SHPRH gene induces apoptosis in neuroblastoma cells

Jingjing Gao[1,*], Hong Pan[1,*], Jie Li[1], Jun Jiang[2] and Wenxian Wang[1]

[1] Department of Nutrition, Shanghai Children's Hospital, School of Medicine, Shanghai Jiao Tong University, Shanghai, China
[2] Endoscopy Center, Minhang District Central Hospital of Fudan University, Shanghai, China
* These authors contributed equally to this work.

## ABSTRACT

**Background:** Circular RNAs (circRNAs) and their derived peptides represent largely unchartered areas in cellular biology, with many potential roles yet to be discovered. This study aimed to elucidate the role and molecular interactions of circSHPRH and its peptide derivative SHPRH-146aa in the pathogenesis of neuroblastoma (NB).

**Methods:** NB samples in the GSE102285 dataset were analyzed to measure circSHPRH expression, followed by *in vitro* experiments for validation. The role of SHPRH-146aa in NB cell proliferation, migration, and invasion was then examined, and luciferase activity assay was performed after SHPRH-146aa and RUNX1 transfection. Finally, the regulation of NB cell apoptosis by SHPRH-146aa combined with NFKBIA was tested.

**Results:** The GSE102285 dataset indicated overexpression of circSHPRH in NB samples, further supported by *in vitro* findings. Overexpression of circ-SHPRH and SHPRH-146aa inhibited proliferation, migration, and invasion of NB cells. A significant increase in apoptosis was observed, with upregulation of Caspase-3 and downregulation of Bcl-2. Furthermore, the peptide derivative SHPRH-146aa, derived from circSHPRH, suppressed NB cell malignancy traits, suggesting its role as a therapeutic target. A direct interaction between SHPRH-146aa and the transcription factor RUNX1 was identified, subsequently leading to increased NFKBIA expression. Notably, NFKBIA knockdown inhibited the pro-apoptotic effect of SHPRH-146aa on NB cells.

**Conclusion:** The study demonstrates that circ-SHPRH and SHPRH-146aa play significant roles in inhibiting the malignant progression of NB. They induce apoptosis primarily by modulating key apoptotic proteins Caspase-3 and Bcl-2, a process that appears to be regulated by NFKBIA. The SHPRH-146aa-RUNX1 interaction further elucidates a novel pathway in the regulation of apoptosis in NB. These findings indicate that circ-SHPRH and its derived peptide SHPRH-146aa could be potential therapeutic targets for NB treatment.

Corresponding authors
Jingjing Gao, gaojj0123@163.com
Wenxian Wang,
wency_2005@hotmail.com

## INTRODUCTION

Neuroblastoma (NB), an embryonic neural crest-derived neoplasm (*Tomolonis, Agarwal & Shohet, 2018*), is a predominant solid malignancy among pediatric cancers (*Khandelwal et al., 2017*; *Patterson, Shohet & Kim, 2011*) accounting for about 8–10% of all cases (*Stermann et al., 2008*). Treatment advancements have unfortunately not significantly improved prognosis, particularly for those in high-risk groups, with 5-year survival rates rarely surpassing 50% (*Park et al., 2013*). Traditional treatment strategies for NB, including surgical resection, chemotherapy, radiation therapy, and immunotherapy, offer a degree of optimism in controlling the disease (*Yelle et al., 2018*). However, it is regrettable that these therapies typically result in severe morbidity, mortality, or complications, which have a detrimental influence on both long-term survival and the quality of life of survivors (*Friedman et al., 2021*). Present research endeavors pivot towards personalized medicine and targeted therapies, capitalizing on genomic insights to anticipate disease progression and treatment response (*Aguilar-Mahecha et al., 2021*). An in-depth comprehension of the intricate molecular mechanisms governing NB could indisputably provide valuable opportunities for enhancing treatment efficacy and patient survival outcomes.

Circular RNAs (circRNAs) have emerged as pivotal regulators in the oncogenic landscape (*Zang et al., 2020*), and through a comprehensive analysis of existing literature and databases, the current circR2Cancer database includes 1,439 associations between 1,135 circRNAs and 82 cancers (*Lan et al., 2020*), including NB, representing a novel research frontier. These covalently closed-loop structured non-coding RNAs play a significant role in modulating gene expression (*Zhang, Yang & Xiao, 2018*), thereby contributing substantially to tumorigenesis and cancer progression (*Huo et al., 2017*). Particularly, circSHPRH has been spotlighted for its potential role in tumorigenesis (*Xiong et al., 2023*). As an exon-intronic circRNA, derived from the SHPRH gene (*Lei et al., 2020*), it is thought to orchestrate a network of genetic interactions influencing cell proliferation and apoptosis, presenting as a potential therapeutic target (*Tran et al., 2022*; *Xiong et al., 2023*). A study had shown that overexpression of SHPRH-146aa in glioblastoma cells reduces its malignant behavior and tumorigenicity *in vitro* and *in vivo* (*Zhang et al., 2018*). Prolonged survival was observed in glioblastoma patients with elevated SHPRH-146aa levels (*Begum et al., 2018*). However, the functional mechanisms of circSHPRH in NB are not fully deciphered, necessitating further exploration (*Mahmoudi et al., 2019*). In parallel, peptides have gained attention for their potential in cancer therapeutics (*Jiang et al., 2011*). As biologically active small molecules, peptides can bind to a variety of cellular targets and affect signaling pathways, thereby opening a promising avenue for targeted cancer therapy (*Moktan & Raucher, 2010*). Nonetheless, their roles in NB and their regulatory effect on circRNAs deserve further examination.

With an emphasis on the circSHPRH in particular, the current work intends to delve deeper into the molecular roles of circRNAs in NB in light of these significant scientific queries and clinical implications. We performed bioinformatics analysis on the NB data set and then performed cell testing to evaluate the impact of circSHPRH overexpression on the malignant behavior of NB cells. Additionally, we investigated the possible correlations

between the transcription factor RUNX1 and the peptide SHPRH-146aa, which is generated from circSHPRH, and the ensuing effects on the expression of NFKBIA. Through these analyses, this study aimed to elucidate the complex interactions between SHPRH-146aa, NFKBIA, and apoptosis regulatory genes, thereby providing new insights into the complex mechanisms of NB pathogenesis.

## MATERIALS AND METHODS

### Expression analysis of circSHPRH in NB

The GSE102285 dataset used in this study was downloaded from the Gene Expression Omnibus (GEO) database (http://www.ncbi.nlm.nih.gov/geo), including four olfactory NB patient samples and one healthy neural control sample. We examined the expression of circSHPRH across these sample types to evaluate its role in NB pathogenesis. Statistical significance was assigned for results where $p < 0.05$.

### Cell lines and culture

By the protocols outlined in the study by *Arnaud-Sampaio et al. (2022)*, we cultured the neuroblastoma cell lines IMR-32, SK-N-SH, SH-SY5Y, and SK-N-AS, along with the human neuron cell line HCN-2. All of these cell lines were obtained from Shanghai Bioleaf Biotech (Shanghai, China). The culture medium comprised RPMI-1640 medium (cat# 30118844; Solarbio, Beijing, China), supplemented with 10% fetal bovine serum (cat# 13011-8611; FBS, Huzhou, China) and 1% penicillin/streptomycin (cat# 15070063; Thermo Fisher Scientific, Waltham, MA, USA) from Zhejiang Tianhang Biotechnology Co., Thermo Fisher Scientific, Waltham, Massachusetts, USA. Cultures were maintained at a constant temperature of 37 °C and a $CO_2$ concentration of 5%.

### Cell transfection

Transfections were performed on cells using either the pcDNA3.1-circSHPRH or the pcDNA3.1-circSHPRH-derived peptide (-146aa) plasmids to overexpress circSHPRH and the peptide, respectively. For knockdown experiments, cells were transfected with siRNA targeting NFKBIA (si-NFKBIA). Empty pcDNA3.1 vector (Invitrogen, Carlsbad, CA, USA) and non-targeting siRNA (si-NC) served as controls for the overexpression and knockdown experiments, respectively. Lipofectamine 3000 (Invitrogen, Carlsbad, CA, USA) was employed for all transfection procedures, the incubation time after transfection was 48 h, adhering strictly to the guidelines provided by the manufacturer.

### Quantitative real-time PCR

We isolated total RNA from NB cell lines with TRIzol reagent (Invitrogen, Carlsbad, CA, USA), adhering to the supplier's protocol. The NanoDrop spectrophotometer (Thermo Fisher Scientific, Waltham, MA, USA) was utilized for RNA quality and concentration assessments. The isolated RNA was then transcribed into complementary DNA (cDNA) by employing a reverse transcription kit (Applied Biosystems, Waltham, MA, USA). This cDNA was then used in subsequent PCR reactions, which were performed using SYBR Green Master (Roche, Basel, Switzerland). Primer sequences employed in the PCR

reactions are detailed in Table S1. Gene expression levels were quantified using the $2^{-\Delta\Delta Ct}$ method, with GAPDH serving as an internal control for normalization.

## Protein extraction and western blotting assay

RIPA Lysis Buffer extracts proteins from cells and the concentration is measured by the BCA protein assay. Proteins were separated by SDS-PAGE and transferred to PVDF membranes. Membranes were incubated with particular primary antibodies (anti-Bcl-2; anti-Caspase-3; anti-SHPRH-146aa; anti-NFKBIA; anti-GAPDH) overnight at 4 °C after being blocked with 5% skim milk in TBST for 2 h at room temperature. These antibodies were used at Bcl-2 (1:1,000; Cell Signaling Technology, Danvers, MA, USA), Caspase-3 (1:1,000; Cell Signaling Technology, Danvers, MA, USA), NFKBIA (1:1,000; Cell Signaling Technology, Danvers, MA, USA) and GAPDH (1:5,000; Cell Signaling Technology, Danvers, MA, USA). After thorough TBST washings, the membranes were incubated with secondary antibodies conjugated to horseradish peroxidase (1:2,000; Cell Signaling Technology, Danvers, MA, USA) for an hour at room temperature, after which protein bands were unveiled *via* ECL technology.

## Assay for cell proliferation

In 96-well plates, cells were seeded at a density of $2 \times 10^3$ cells/well and incubated for 24, 48, 72, and 96 h, respectively. Each well was then filled with 10 μl of Cell Counting Kit-8 (CCK-8) solution before being incubated at 37 °C for a further 2 h. Absorbance was subsequently measured at 450 nm using a microplate reader.

## Assays for cell migration and invasion

In migration assays, we loaded $3 \times 10^4$ cells, cultured in serum-free media, onto the upper chamber of a Transwell insert (8 μm pore size; Corning, Corning, NY, USA). For invasion tests, Transwell membranes were pre-coated with Matrigel (BD Biosciences, Franklin Lakes, NJ, USA), followed by the addition of $1 \times 10^5$ cells to the upper chamber. The lower chamber contained medium supplemented with 10% FBS, serving as a chemoattractant. After a 24-h incubation period, cells that migrated or invaded through the membrane were stained with 0.1% crystal violet and subsequently counted using a microscope.

## Apoptosis analysis

Apoptosis was assessed by Flow Cytometry Apoptosis Detection Kit (BD Biosciences, Franklin Lakes, NJ, USA). Briefly, harvested cells were washed with cold PBS and resuspended in the provided binding buffer. Following this, cells were subjected to a 15-min staining process with Annexin V-FITC and propidium iodide (PI) at room temperature, ensuring the process was conducted in darkness. Following the staining procedure, flow cytometry was employed to analyze the apoptotic rates in the cells.

## Assay for luciferase reporter

Adhering to the Dual-Luciferase Reporter Assay System (Promega, Madison, WI, USA) instructions, NB cells were co-transfected with the Renilla luciferase reporter plasmid and either the circSHPRH peptide overexpression plasmid or control vector. Alongside this,

luciferase reporters carrying NFKBIA promoter sequences (wild-type or mutant) were also co-transfected into the cells. Following 2 days post-transfection, dual-luciferase assay was carried out, with Firefly and Renilla luciferase activities measured using a microplate reader (BioTek, Winooski, VT, USA).

## Co-immunoprecipitation assay

Cells transfected with either an empty vector or vectors expressing CircSHPRH-flag were lysed using the provided IP lysis buffer. The resultant cell lysates were then incubated with either specific antibodies for the respective transfected proteins or control IgG at 4 °C overnight with gentle rotation. Subsequently, Protein A/G agarose beads were utilized to capture the antibody-protein complexes. After washing to remove unbound proteins, the immunoprecipitated proteins were eluted from the beads and analyzed by WB for specific protein interactions.

## Statistical analysis

All experiments in this study were conducted independently and in triplicate. Data are shown as mean ± standard deviation (SD). The Student's unpaired t-test was utilized for evaluating significant differences between groups. A $p$-value of less than 0.05, denoted with an asterisk "*", was considered indicative of statistical significance. The statistical methodology employed in this study was informed by the approach detailed in *Tang et al. (2023)*.

# RESULTS

## Overexpression of circ-SHPRH suppresses NB progression

The GSE102285 dataset revealed a notable low expression of circ-SHPRH in NB samples (Fig. 1A). Further exploration into circ-SHPRH demonstrated a substantial decrease in expression within NB cell lines when compared to human neuronal cells (HCN-2) (Fig. 1B), with a distinct prominence in SK-N-AS and SH-SY5Y cell lines. The circ-SHPRH overexpression vector was then transfected into these two cells. Following transfection, its efficiency was evaluated *via* qRT-PCR, and a significant increase in circ-SHPRH expression confirmed successful transfection (Fig. 1C). The CCK-8 experiment revealed that circ-SHPRH overexpression dramatically reduced cell proliferation (Figs. 1D and 1E). A Transwell experiment was also revealed that cells overexpressing circ-SHPRH had considerably lower migratory and invasive capacities (Figs. 1F–1H).

## circ-SHPRH induces apoptosis in NB cells

Our study demonstrated a significant increment in the proportion of apoptotic cells upon circ-SHPRH overexpression compared to the control group, with late apoptosis rates escalating from 23.0% to 33.2% and 20.7% to 25.5% in respective groups, as illustrated in Figs. 2A–2C. This observation suggests a potential pro-apoptotic role of circ-SHPRH in NB. Next, we analyzed the expression of Bcl-2 and Caspase-3, two apoptosis-related proteins. Among them, the Bcl-2 protein is an important part of the apoptosis mechanism,

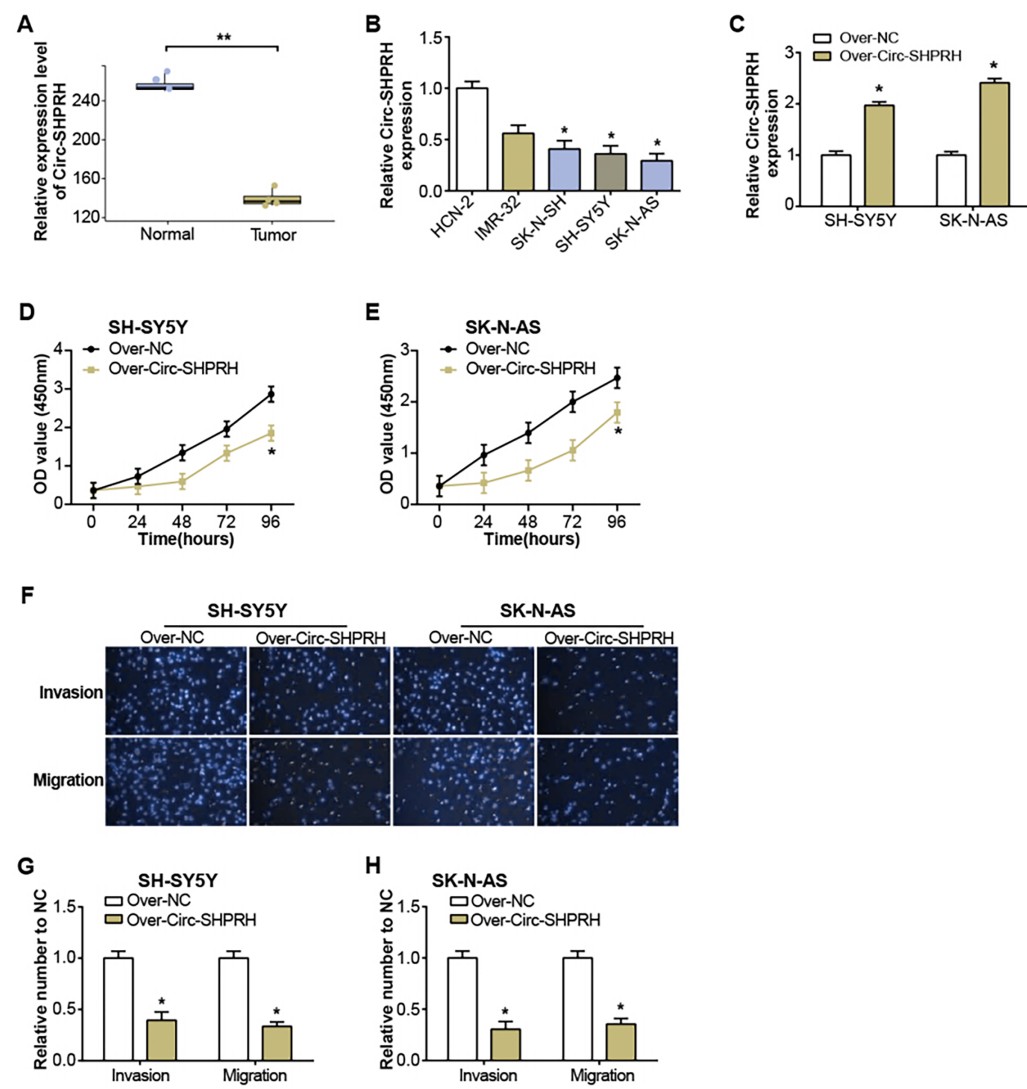

**Figure 1 Expression of circ-SHPRH and its associated effect on NB cell phenotype.** (A) circ-SHPRH expression in the GSE102285 dataset, with the blue box line representing normal samples and the orange box line representing tumor samples. (B) qRT-PCR analysis of circ-SHPRH mRNA levels in NB cell lines. (C) qRT-PCR detection of circ-SHPRH expression levels after overexpression transfection of SH-SY5Y and SK-N-AS cells. (D and E) CCK-8 assay examines the regulation of circ-SHPRH overexpression on the proliferation of SH-SY5Y and SK-N-AS cells. (F–H) Transwell assay to analyze the migration and invasion abilities of SH-SY5Y and SK-N-AS cells after circ-SHPRH overexpression. $^{*}P < 0.05$, $^{**}P < 0.01$.

and its main function is to inhibit apoptosis. Conversely, as an essential effector in the apoptotic pathway, Caspase-3 fosters cellular demise. Our qRT-PCR and WB results revealed contrasting expression patterns of these apoptotic factors in response to circ-SHPRH overexpression. Specifically, Bcl-2 expression was downregulated, while Caspase-3 expression was upregulated (Figs. 2D–2F). These observations suggest that the enhanced expression of circ-SHPRH may tip the balance of apoptosis regulation in NB cells. Consequently, these findings underscore the potential role of circ-SHPRH in facilitating apoptosis, thereby exhibiting a tumor-suppressive effect in NB.

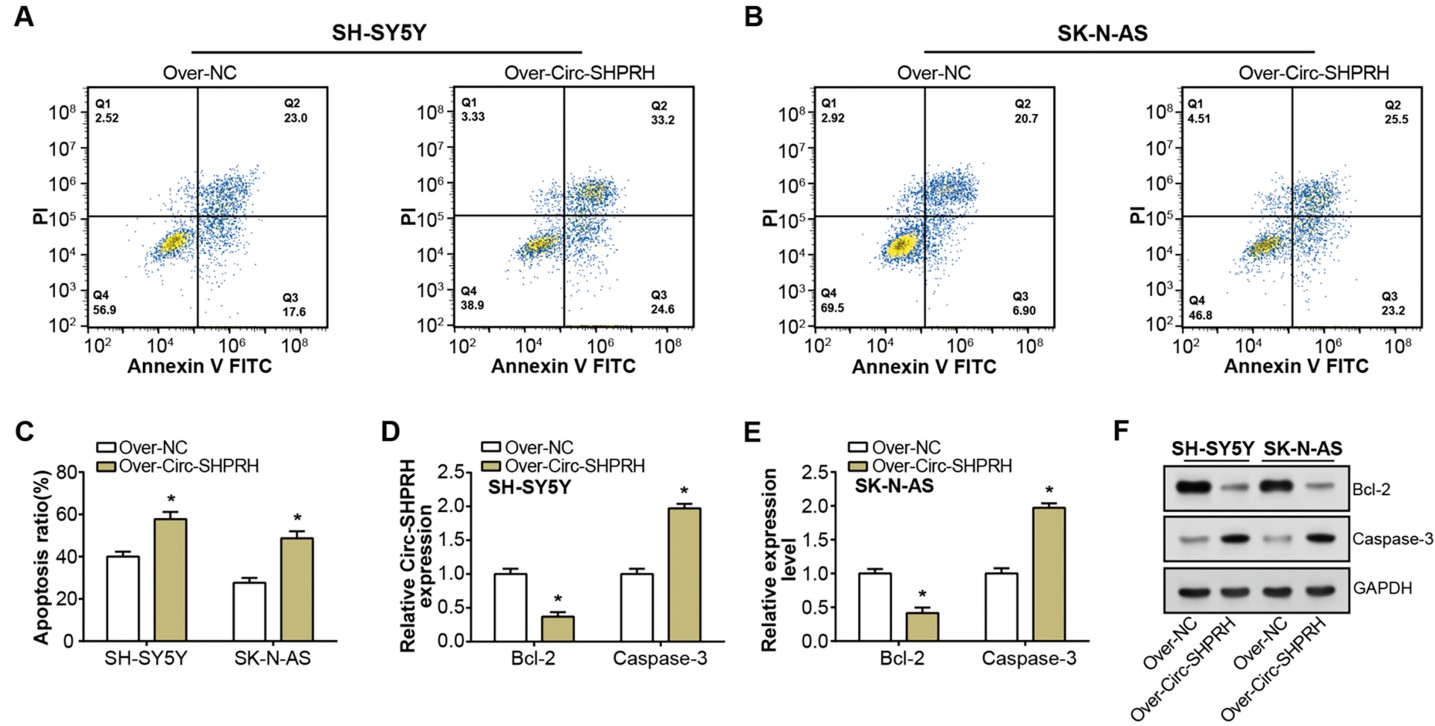

**Figure 2 Effect of circ-SHPRH overexpression on NB cell apoptosis.** (A and B) Flow cytometry analysis evaluating the cell apoptosis rate in neuroblastoma cell lines upon circ-SHPRH overexpression. The four quadrants represent viable cells (Q4), early apoptotic cells (Q3), late apoptotic cells (Q2), and necrotic cells (Q1), respectively. (C) Quantitative representation of flow cytometry results showing enhanced apoptosis after circ-SHPRH overexpression. (D and E) qRT-PCR analysis of Bcl-2, Caspase-3 expression changes after circ-SHPRH overexpression in SH-SY5Y and SK-N-AS cells. (F) WB analysis of changes in expression of Bcl-2 and Caspase-3 proteins after circ-SHPRH overexpression. *$P < 0.05$.

## Overexpression of SHPRH-146aa inhibits the malignant progression of NB

The SHPRH-146aa is a peptide derived from the circular RNA, circ-SHPRH. *Zhang et al. (2018)* have previously identified this peptide as a novel tumor suppressor protein, SHPRH-146aa, noting its influence on the survival of glioblastoma patients. Motivated by this finding, we sought to study the effect of circ-SHPRH-146aa on the pathophysiology of NB, prompting us to engineer its overexpression in NB cells (Fig. 3A). The consequent *in vitro* experiments demonstrated that the overexpression of SHPRH-146aa significantly inhibited the malignant characteristics of SH-SY5Y and SK-N-AS cells, including proliferation, migration, and invasion activities (Figs. 3B–3F). These findings indicated that SHPRH-146aa exerts an inhibitory effect on the progression of NB, and provided more evidence to support SHPRH-146aa as a potential therapeutic target.

## circ-SHPRH peptide-RUNX1 interaction augments NFKBIA expression in NB

To assess the overexpression efficiency of SHPRH-146aa in SH-SY5Y and SK-N-AS cell lines, we performed qRT-PCR analysis (Fig. 4A) and WB analysis (Fig. 4B). The results confirmed that SHPRH-146aa was successfully overexpressed in both cell lines. Co-IP

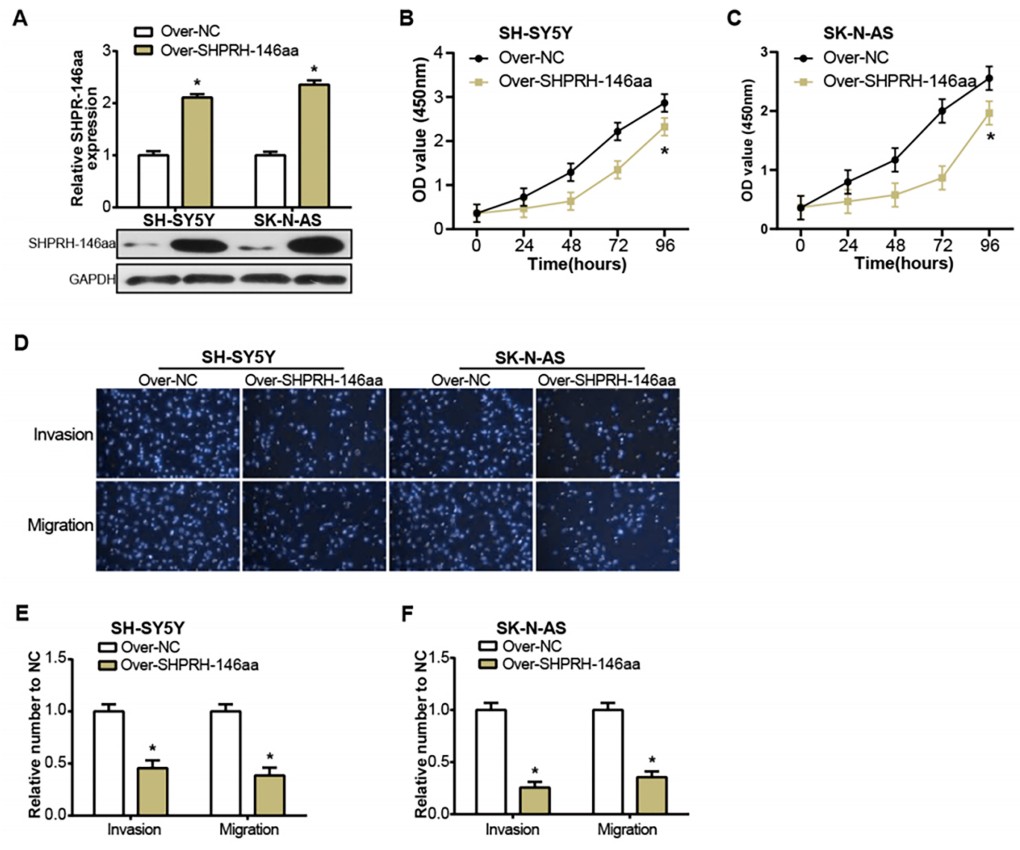

**Figure 3 Regulation of NB cells by overexpression of circSHPRH-146aa.** (A) qRT-PCR and WB analysis of the overexpression efficiency of SHPRH-146aa in SH-SY5Y and SK-N-AS cell lines. (B and C) CCK-8 assay analysis of cell proliferation in SH-SY5Y and SK-N-AS cell lines after SHPRH-146aa overexpression. (D–F) Transwell assay assessing the effect of SHPRH-146aa overexpression on cell migration and invasion abilities in SH-SY5Y and SK-N-AS cell lines. $^*P < 0.05$.

assays indicated a direct interaction between SHPRH-146aa and RUNX1, as evidenced by the co-precipitation of these proteins (Figs. 4C and 4D). Previous studies have shown a role for RUNX1 in binding and regulating the NFKBIA promoter region. In line with this, our study demonstrated, through qRT-PCR and WB, that SHPRH-146aa overexpression significantly increased NFKBIA expression in neuroblastoma cells (Figs. 5A and 5B). This finding implies that SHPRH-146aa may modulate NFKBIA expression *via* direct interaction with RUNX1. Further analysis using JASPAR (http://jaspar.genereg.net) established the binding profile between RUNX1 and NFKBIA (Fig. 5C). Luciferase assays showed enhanced activity in cells transfected with the wild-type (WT) plasmid, while mutant (MUT) plasmid-transfected cells exhibited reduced activity, indicating mutations affecting promoter function (Fig. 5D). Transfection with SHPRH-146aa peptide resulted in significantly increased luciferase activity, suggesting it enhances promoter activity and target gene transcription. Co-transfection with SHPRH-146aa and RUNX1 further augmented this activity, underscoring their crucial roles in the regulation of promoter activity and gene transcription (Fig. 5E).

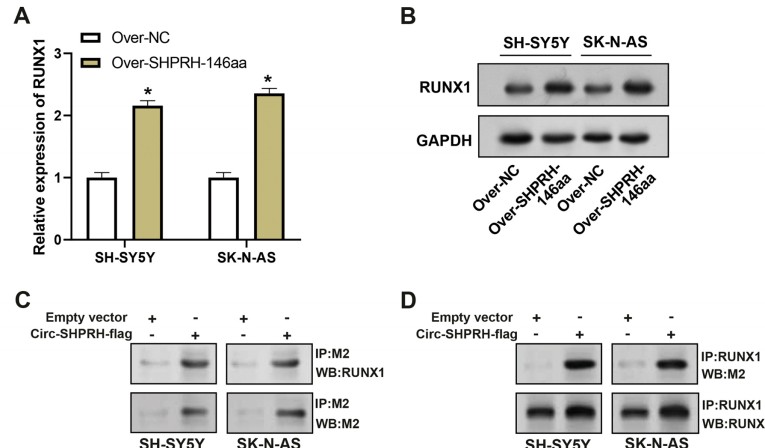

**Figure 4 Co-IP analysis of the interaction between circSHPRH-146aa and RUNX1.** (A) qRT-PCR analysis of the overexpression efficiency of SHPRH-146aa in SH-SY5Y and SK-N-AS cell lines. $*P < 0.05$. (B) WB analysis of the overexpression efficiency of SHPRH-146aa in SH-SY5Y and SK-N-AS cell lines. (C and D) WB images illustrate the presence of RUNX1 in immunoprecipitates, confirming the interaction between SHPRH-146aa and RUNX1. IP: Co-immunoprecipitation, WB: Western blotting, "+" indicates that the condition is added, "−" indicates that the condition is not added, and the gray value of the band reflects the protein level after treatment with different conditions.

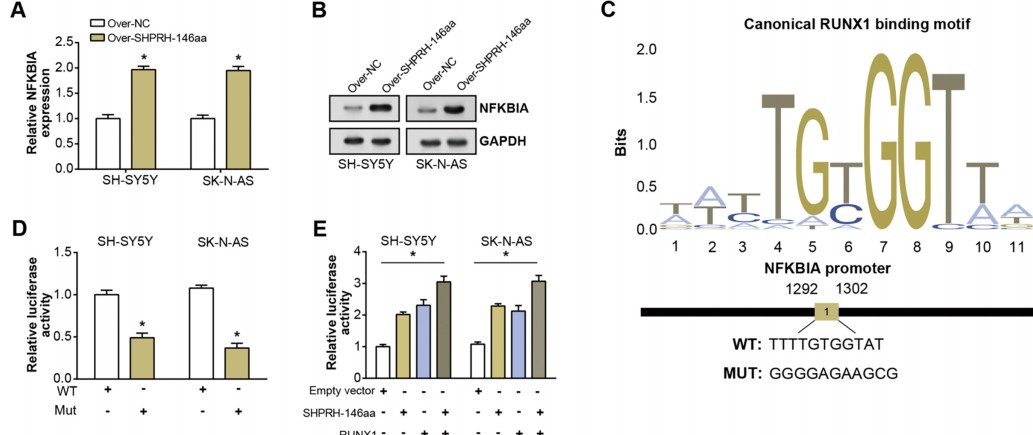

**Figure 5 Regulation of circSHPRH-146aa and RUNX1 in promoter activity and gene transcription.** (A) qRT-PCR detection of the expression level of NFKBIA after SHPRH-146aa overexpression in SH-SY5Y and SK-N-AS cells. (B) WB detection of NFKBIA expression level after SHPRH-146aa overexpression in SH-SY5Y and SK-N-AS cells. (C) The RUNX1 binding motif and its corresponding site on the NFKBIA promoter. (D and E) Bar graphs, the left panel shows the effect of wild-type (WT) and mutant (MUT) plasmids on luciferase activity, and the right panel shows the effect of SHPRH-146aa and RUNX1 on promoter activity. The X-axis represents the respective conditions and the Y-axis represents the relative luciferase activity. $*P < 0.05$.

## NFKBIA silencing attenuates the pro-apoptotic action of SHPRH-146aa in NB cells

Flow cytometry analysis on NB cell lines showed an elevated rate of apoptosis following overexpression of the SHPRH-146aa. However, this increase in apoptosis was partly nullified
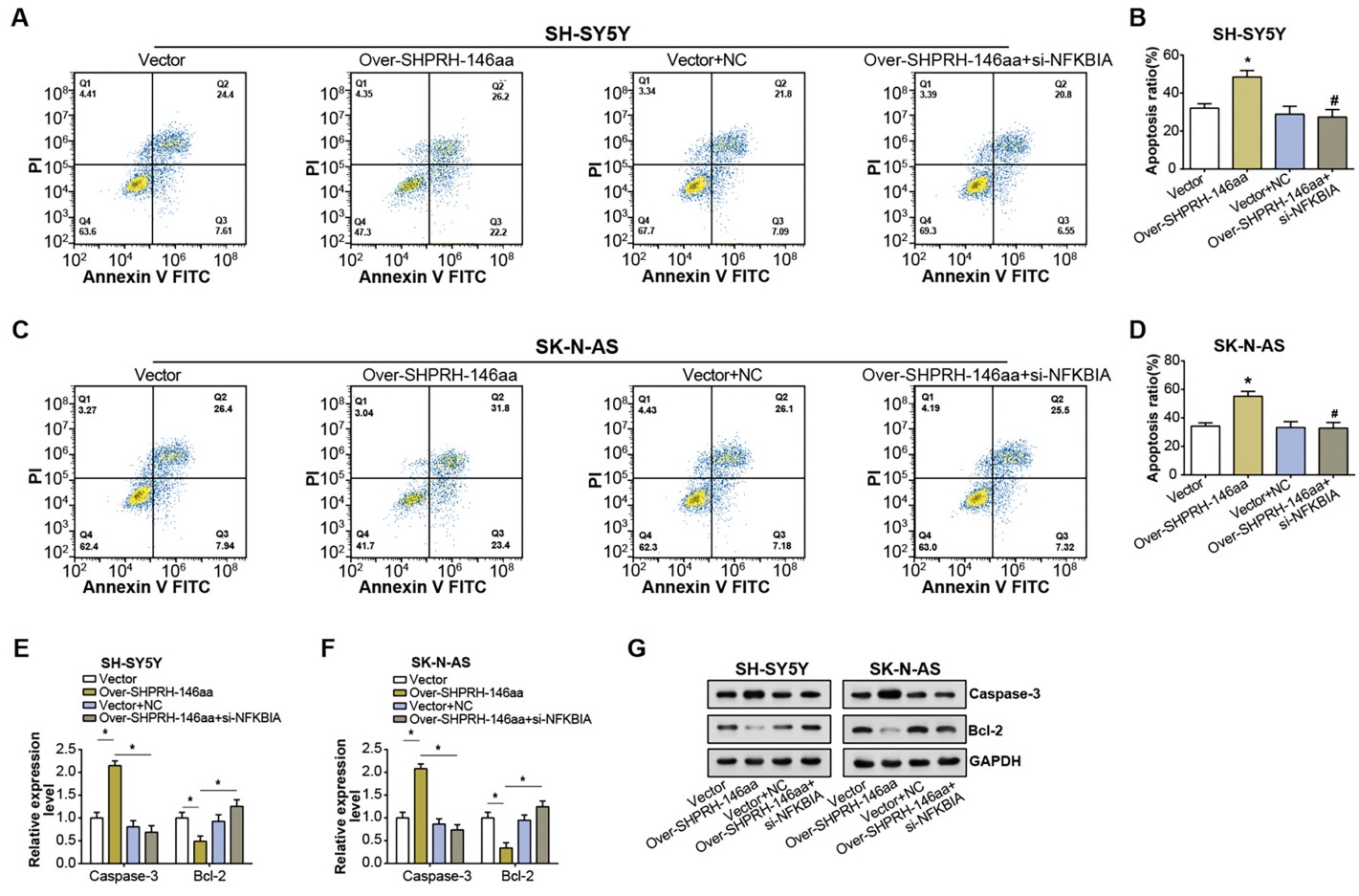

**Figure 6 Effect of circSHPRH-146aa overexpression and NFKBIA knockdown on NB cell apoptosis.** (A–D) Flow cytometry analysis of apoptosis in NB cell lines under different conditions: vector, over-SHPRH-146aa, vector+NC, over-SHPRH-146aa+si-NFKBIA. The four quadrants in the flow cytometry plot represent different cell populations based on Annexin V and propidium iodide staining, indicating early apoptotic, late apoptotic, live, and dead cells, respectively. (E and F) qRT-PCR analysis detected the expression of Bcl-2 and Caspase-3 in NB lines after SHPRH-146aa overexpression and NFKBIA knockdown. (G) WB analysis to detect the expression of Bcl-2 and Caspase-3 in NB lines after SHPRH-146aa overexpression and NFKBIA knockdown. $^{*}P < 0.05$, $^{\#}P < 0.05$ *vs.* Over-SHPRH-146aa group. 

by si-NFKBIA, suggesting a regulatory role of NFKBIA in the SHPRH-146aa-induced apoptosis (Figs. 6A–6D). Further evaluation of apoptosis-associated gene expression *via* qRT-PCR and WB assays revealed significant changes following the overexpression of the SHPRH-146aa. Specifically, we observed an upregulation of Caspase-3 and a downregulation of Bcl-2. In contrast, the changes in apoptosis-related genes expression induced by SHPRH-146aa overexpression were attenuated after si-NFKBIA co-transfection (Figs. 6E–6G). These findings suggest that the SHPRH-146aa may promote apoptosis in NB cells through modulation of Bcl-2 and Caspase-3, a process possibly dependent on NFKBIA. This suggests that a potential regulatory circuit involving SHPRH-146aa, NFKBIA, and apoptosis-related genes is involved in the pathogenesis of NB.

## DISCUSSION

The field of neuroblastoma research has seen a growing interest in the involvement of circRNAs, which are emerging as critical players in tumor biology (*Klironomos et al., 2019*). It was demonstrated by *Li et al. (2019a)* that the circ-CUX1/EWSR1/MAZ axis presents a viable candidate for therapy by hindering glycolysis and neuroblastoma development. Correspondingly, circ0125803 was reported to promote NB progression through the absorption of miR-197-5p and the subsequent enhancement of E2F1 expression (*Tang et al., 2022*). Another piece of literature revealed a role for circ-CUX1 in accelerating neuroblastoma progression through the miR-16-5p/DMRT2 pathway (*Zhang et al., 2020*). Moreover, circKIF2A has been implicated in neuroblastoma development by modulating PRPS1 expression through the suppression of miR-377-3p (*Jin et al., 2022*). Collectively, these studies point to the complex functions that circRNAs play in the development of NB and emphasize their promise as candidate targets. Nevertheless, the precise mechanistic interplay between various circRNAs and NB pathogenesis continues to be an area ripe for further exploration.

The potential therapeutic utility of SHPRH-146aa, a peptide derived from the circular RNA circ-SHPRH, has emerged as a notable finding from our study. In the context of NB models, SHPRH-146aa exhibited a notable inhibitory effect on malignant cell behaviors, underscoring its potential as a therapeutic agent in halting neuroblastoma progression. A key discovery of our study is the direct interaction between SHPRH-146aa and the RUNX1 transcription factor, a mechanism suggesting the possibility of modulating SHPRH-146aa activity through co-regulatory pathways. This interaction underscores the intricate nature of circRNA-derived peptides in cellular dynamics. RUNX1, also known as Runt-related transcription factor 1, plays a crucial role in hematopoiesis and bone development. The research by *Li et al. (2019b)* confirmed the function of RUNX1 in promoting colorectal cancer metastasis, and its upregulation can promote cell metastasis and epithelial-mesenchymal transition. Furthermore, *Hong et al. (2019)* highlighted the critical function of RUNX1 in normal cell lineages and tissue development, pointing out its dual role as an oncogene or tumor suppressor in solid tumors, particularly in breast cancer, where it suppresses invasiveness and Affect epithelial-mesenchymal transition. The association of RUNX1 with breast cancer metastasis has also been confirmed in other studies (*Ellis et al., 2012*; *Heilmann, 2017*). Given the recognized role of RUNX1 in a multitude of physiological and pathological contexts, the interaction between SHPRH-146aa and RUNX1 potentially signifies a novel regulatory axis in NB.

Our study has advanced the understanding of the roles played by circ-SHPRH and its derivative peptide, SHPRH-146aa, in the pathogenesis of NB. The complex interaction between SHPRH-146aa and RUNX1 was found to notably augment the expression of NFKBIA in NB. NFKBIA, also known as IκBα, acts as an inhibitor of the NF-κB transcription factor, which performs a vital function in inflammatory and immune responses (*De Hong et al., 2015*; *Hellweg, 2015*). Notably, dysregulation in the NF-κB pathway has been implicated in various cancers. Prior studies have identified a tumor-suppressive role of NFKBIA in Hodgkin's lymphoma (*Cabannes et al., 1999*),

colorectal cancer (*Gao et al., 2007*), and hepatocellular carcinoma (*He et al., 2009*) for specific single nucleotide polymorphisms and haplotypes of NFKBIA (*Shen et al., 2015*). These findings underscore NFKBIA as a potential tumor suppressor and highlight the importance of exploring its role in cancer development. This relationship, confirmed by our Co-IP results and further substantiated by qRT-PCR and WB assays, proposes a new mechanism through which SHPRH-146aa modulates NFKBIA expression *via* direct interaction with RUNX1. Further examination of the transcriptional activity of RUNX1 and NFKBIA revealed the critical roles these two factors play in regulating promoter activity and gene transcription. In addition to the effects of SHPRH-146aa on NFKBIA expression, our results indicate that NFKBIA is a key regulator of circ-SHPRH-mediated apoptosis in NB cells. Flow cytometry demonstrated that SHPRH-146aa overexpression increases apoptosis, an effect attenuated by NFKBIA knockdown. The observed alterations in the expression of apoptosis-associated genes, such as Bcl-2 and Caspase-3, following SHPRH-146aa overexpression, indicate a potential regulatory network involving SHPRH-146aa, NFKBIA, and apoptosis-related genes. This intricate regulatory circuit presents new avenues for research into the complex dynamics underlying NB pathogenesis.

## CONCLUSION

In conclusion, this study elucidates a novel role of circ-SHPRH and its derived peptide SHPRH-146aa in the pathogenesis of NB. We observed that SHPRH-146aa exerts an inhibitory effect on NB cell proliferation, migration, and invasion, implying its potential as a therapeutic target. Further, our investigations unveiled a regulatory interaction between SHPRH-146aa and the RUNX1 transcription factor, leading to enhanced NFKBIA expression. Significantly, NFKBIA was shown to modulate circ-SHPRH-induced apoptosis in NB cells, providing insights into a complex regulatory circuit involving circ-SHPRH, NFKBIA, and apoptosis-related genes. These findings emphasize the significance of circRNAs and their peptides in cancer biology, uncovering new avenues for neuroblastoma research while offering potential therapeutic strategies.

## ACKNOWLEDGEMENTS

The authors thank all patients involved in this study.

### Funding

The authors received no funding for this work.

### Competing Interests

The authors declare that they have no competing interests.

### Author Contributions

- Jingjing Gao analyzed the data, prepared figures and/or tables, and approved the final draft.

- Hong Pan conceived and designed the experiments, prepared figures and/or tables, and approved the final draft.
- Jie Li performed the experiments, analyzed the data, authored or reviewed drafts of the article, and approved the final draft.
- Jun Jiang analyzed the data, prepared figures and/or tables, and approved the final draft.
- Wenxian Wang conceived and designed the experiments, authored or reviewed drafts of the article, and approved the final draft.

## Microarray Data Deposition

The following information was supplied regarding the deposition of microarray data:

The transcriptome analysis is available at GEO: GSE102285.

## Data Availability

The raw measurements are available in the Supplemental Files.

## Supplemental Information

Supplemental information for this article can be found online at http://dx.doi.org/10.7717/peerj.16806#supplemental-information.

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
