# Peer review of "A peptide encoded by the circular form of the SHPRH gene induces apoptosis in neuroblastoma cells"

_PeerJ, doi:10.7717/peerj.16806_

## Round 0.1 · original submission · Major Revisions

The paper needs major revision. Please follow the indications given by the reviewers and address all points raised.

Reviewer 1 ·

Basic reporting

The study by Wang et al. deals with the role of NFKBIA as a key regulator of circSHPRH-induced apoptosis in NB cells. The topic is interesting even if there are some issues that need to be considered by the authors.

Experimental design

NB is usually located in the adrenal gland or along the spinal ganglia. It would be better to specify the location of the tumors analyzed and compare the data with the healthy tissue of the same organ/site.

It would be more appropriate to conduct studies on MYCN amplified NB cells which are the most malignant, since MYCN amplification is a recognized negative prognostic factor of NB.

Validity of the findings

As the experiments have been carried out by using NB cells without MYCN amplification, the potential therapeutic role of this peptide is extremely limited.

Additional comments

Line 211: Considering that SHPRH has several targets, what are the reasons why only Co-IP with RUNX1 was analyzed?
Lines 232-236: Since SHPRH-146aa overexpression induces Bcl-2 upregulation and caspase 3 downregulation, it plays an anti-apoptotic and not a pro-apoptotic role.

Reviewer 2 ·

Basic reporting

The manuscript could benefit from language editing to enhance clarity. Some specific points for potential improvements in basic reporting are as follows:
- Line 186-187: “Our findings revealed a marked increase in apoptotic cell percentages following circSHPRH overexpression relative to the control group, late-apoptosis rising from 23.0% to 33.2% and from 20.7% to 25.5% in the respective groups (Figures 2A-2C)”. The sentence appears to be describing the quantified values in Figure 2A-B, with each panel representing a cell line, SH-SY5Y or SK-N-AS. The use of the word “respectively” appears to suggest one cell line is the control, which is not the case in this context. Furthermore, these values presented in the text appear to have come from one replicate. It may be more representative to use values in texts that are representative of multiple replicates to ensure reproducibility.
- For all figure captions, it is recommended to indicate the number of biological replicates or independent experiments performed, and what the error bars represent (if applicable).
- For overexpression or siRNA experiments involving transfection, due to the transient nature of the method, it is recommended to include more information about how long after the transfection the experiment was conducted and the phenotype observed. This could preferably be included in the figure captions.
- It is recommended to improve the transition to investigating NFKBIA (starting line 226) and provide more introduction into the rationale for looking at NFKBIA.
- References should be adjusted to PeerJ style.

Experimental design

The experimental approaches used in this manuscript appear feasible but could benefit from additional clarifications in describing the techniques in the methods section. Specific comments are as follows:
- Line 102-104 “Transfections were performed on cells using either the pcDNA3.1-circSHPRH or the pcDNA3.1-circSHPRH-derived peptide (-146aa) plasmids to overexpress circSHPRH and the peptide, respectively” – it wasn’t clear what the difference is between the two constructs used. It may be advisable to provide the source/supplier of the construct or a sequence file. This would help with the reproducibility of the work being reported.
- For the description of experimental methods, although the authors made references to other articles for protocols they followed, it is still recommended to summarize the methods performed.
- For all reagents used, particularly the antibodies, it is recommended to provide the specific catalog number from the manufacturer. This would help with the reproducibility of the results.

Validity of the findings

There are some concerns regarding the validity of some findings. Specific comments are as follows:
- The “dramatically reduced cell proliferation” claim, as described in lines 179-180, appears to be a very mild effect instead. The growth rate/doubling time of the cells in either condition, as shown in each panel of Figure 1D-E and Figure 3B-C, appear very similar. Furthermore, there seem to be some discrepancies in seeding on t=0, with slightly lower seeding for the group with over-circSHPRH. It cannot be ruled out that the seeding differences may be the cause of the mild difference in proliferation. It is recommended to perform appropriate normalizations or to repeat the experiment with more precise loading to ensure that the difference in proliferation is representative and reproducible.
- As previously mentioned in the Experimental Design section, the manuscript did not make clear the distinction between the overexpression construct used in Figures 1-2 and that of Figure 3. Although it is expected that the data presented in Figure 3 is consistent with that of Figure 1, there was no validation in Figure 3 to show that the circSHPRH-146aa sequence was indeed expressed in its peptide form. If the circSHPRH-146aa was overexpressed with a FLAG tag, it is recommended to include the validation of overexpression by western blot for the data presented in Figure 3.
- There are a significant number of non-specific bands in the post-IP immunoblot, as shown in the raw data of Figure 4. Although the interpretations appear reasonable, it is recommended to confirm that the bands shown are correct, including labeling the molecular weight of the bands and comparing it to the expected molecular weight.
- Also in Figure 4: it may be informative to confirm the levels of RUNX1 before IP (in cell lysates) to check if RUNX1 levels are affected by circSHPRH-146aa overexpression in cell lysates.

·

Basic reporting

1. In previous studies, circSHPRH was described as circ-SHPRH, circSHPRH-146aa was described as SHPRH-146aa. Please change the description in this article for future publication.
2. The expression of line30-31 is inappropriate , circRNAs and their derived peptides can represent an unknown area in celluar biology?
Circular RNAs (circRNAs) and their derived peptides represent largely unchartered areas in cellular biology.
3. Line35-37 mentioned this study highlights circSHPRH-146aa’s potential as a therapeutic target, please add relevant references and explain the current research progress on this therapeutic target.
Furthermore, our results showed that the overexpression of circSHPRH-146aa impairs NB cell proliferation, migration, and invasion, thus underlining its potential as a therapeutic target.
4. In line57, result in severe morbidity? Or mortality or complications?
However, it is regrettable that these therapies typically result in severe morbidity.
5. In line64,circRNAs have emerged as pivotal regulators in the oncogenic landscape, this contradicts the statement in the first sentence of the abstract. Is there a considerable amount of research on circRNAs? If so, please add relevant references and revise the first paragraph.
Circular RNAs (circRNAs) have emerged as pivotal regulators in the oncogenic landscape
6. In line78-80,the purpose of this study should be to probe the mechanism of circ-SHPRH pathway in the pathogenesis of NB, rather than to further explore the function of circRNAs.
Given these compelling scientific questions and clinical implications, the present study aims to probe deeper into the molecular functions of circRNAs in NB, with a particular focus on the circ-SHPRH.
7. In line173,the GSE102285 dataset revealed a notable overexpression of circ-SHPRH in NB samples, then compared with human nerve cells, the expression of circ-SHPRH in NB cell was reduced, so whether circ-SHPRH was overexpressed or underexpressed in NB.
The GSE102285 dataset revealed a notable overexpression of circSHPRH in NB samples.
8. In line207-208, it should be these results provide additional evidence support for SHPRH-146aa as a potential therapeutic target. At the same time , please add relevant references.
These findings indicate that circSHPRH-146aa exerts an inhibitory effect on the progression of NB, thereby unveiling a potential therapeutic target.
9. The language expression of part of the article is too complicated, especially the discussion part, which is not easy to understand intuitively.
10. Each of the pictures in the figures should be annotated accordingly.
11. Abbreviations that appear for the first time should include the full name (including figures).

Experimental design

no comment

Validity of the findings

no comment

Additional comments

no comment

---

## Round 0.2 · Minor Revisions

The authors have adequately addressed the issues raised by the reviewers. Please make the additional revisions suggested.

Reviewer 2 ·

Basic reporting

The authors have adequately addressed the comments from the previous round of review.
A minor issue: in the title, the word “form” appears misspelled as “from”. It is also suggested to mention SHPRH as a gene, i.e., “… the circular form of the SHPRH gene…” in the title.

Experimental design

no comment

Validity of the findings

no comment

---

## Round 0.3 · accepted · Accept

The authors have adequately addressed the issues raised by the reviewers. The manuscript is now ready for publication.